Spatiotemporal patterns of the COVID-19 epidemic in Mexico at the municipality level

http://orcid.org/0000-0002-6138-9879 Mas Jean-François 1 jfmas@ciga.unam.mx
Pérez-Vega Azucena 2
1 Laboratorio de análisis espacial, Centro de Investigaciones en Geografía Ambiental, Universidad Nacional Autónoma de México , Morelia, Michoacán , Mexico
2 Departamento de Geomática e Hidraúlica, Universidad de Guanajuato , Guanajuato, Guanajuato , Mexico
Aly Sharif
Electronic publication date: 2021 Dec 24
Publication date: 2021
Volume: 9
Electronic Location ID: e12685
Received 2021 Jan 25; Accepted 2021 Dec 3
Copyright: © 2021 Mas and Pérez-Vega
Copyright year: 2021
Copyright holder: Mas and Pérez-Vega
License: This is an open access article distributed under the terms of the Creative Commons Attribution License, which permits unrestricted use, distribution, reproduction and adaptation in any medium and for any purpose provided that it is properly attributed. For attribution, the original author(s), title, publication source (PeerJ) and either DOI or URL of the article must be cited.
License URL: https://creativecommons.org/licenses/by/4.0/

Keywords: SARS-CoV-2 pandemic, COVID-19, Spatial analysis, GIS, Moran’s index, Spatial scan, Cluster, Contagion patterns

Funding: Dirección General de Asuntos del Personal Académico, Universidad Nacional Autónoma de México PAPIME PE117519 This research was supported by the Project PAPIME PE117519 “Herramientas para la enseñanza de la Geomática con programas de código abierto” (Dirección General de Asuntos del Personal Académico, Universidad Nacional Autónoma de México). The funders had no role in study design, data collection and analysis, decision to publish, or preparation of the manuscript.

==============================
In recent history, Coronavirus Disease 2019 (COVID-19) is one of the worst infectious disease outbreaks affecting humanity. The World Health Organization has defined the outbreak of COVID-19 as a pandemic, and the massive growth of the number of infected cases in a short time has caused enormous pressure on medical systems. Mexico surpassed 3.7 million confirmed infections and 285,000 deaths on October 23, 2021. We analysed the spatio-temporal patterns of the COVID-19 epidemic in Mexico using the georeferenced confirmed cases aggregated at the municipality level. We computed weekly Moran’s I index to assess spatial autocorrelation over time and identify clusters of the disease using the “flexibly shaped spatial scan” approach. Finally, we compared Euclidean, cost, resistance distances and gravitational model to select the best-suited approach to predict inter-municipality contagion. We found that COVID-19 pandemic in Mexico is characterised by clusters evolving in space and time as parallel epidemics. The gravitational distance was the best model to predict newly infected municipalities though the predictive power was relatively low and varied over time. This study helps us understand the spread of the epidemic over the Mexican territory and gives insights to model and predict the epidemic behaviour.

Introduction

In December 2019, a novel coronavirus SARS-CoV-2 (severe acute respiratory syndrome coronavirus 2) was first reported in Wuhan, China. Since then, the pandemic of Coronavirus Disease (COVID-19) has spread globally, with more than 242 millions of cases and 4.9 millions of deaths registered worldwide (https://covid19.who.int/). By October 23, 2021, the most affected WHO region by COVID-19 pandemic, with almost 92,641,000 confirmed cases, was the Americas. In this region, the countries that present the most substantial number of deaths were the USA, Brazil and Mexico.

According to the Mexican Health Secretary’s records, the first confirmed cases of COVID-19 in Mexico were reported in February 2020, and community transmission started March 23, 2020. Since this date, COVID-19 has spread over the Mexican territory, with over 3,767,000 accumulated confirmed cases and 285,000 deaths by October 23, 2021 (https://www.gob.mx/salud/documentos/datos-abiertos-152127). Current evidence suggests that the virus is transmitted primarily between people in close contact with each other. COVID-19 is a mostly airborne disease: A person can become infected when aerosols or droplets containing the virus are inhaled or directly contact the eyes, nose, or mouth (Cheng et al., 2020; Ghinai et al., 2020; Greenhalgh et al., 2021; Luo et al., 2020).

In the absence of an effective treatment or a vaccine, the Mexican Government declared a health emergency and implemented a range of sanitary measures to contain the virus spreading. These measures included a national campaign to promote social distancing, called “Jornada Nacional de Sana Distancia” (National Workday of Healthy Distance), an increase of health expenditure and the closing of non-essential economic activities from March 23 to May 30, 2020. After this lockdown period, gradual reactivation of economic activities was initiated and tuned at the state level using colour-coded restriction levels (Acuña-Zegarra, Santana-Cibrian & Velasco-Hernandez, 2020). The restrictions were determined, considering hospital occupancy and its trend and the incidence rates of each state and its neighbours. Schools, colleges and universities suspended classes, and academic activities were carried out remotely. A national vaccination plan against COVID-19 was initiated in December 2020, focussing first on the Health workers dealing with COVID-19 (December 2020–February 2021) and other health workers and the elderly (February–April 2021). At the early stage of the campaign, the vaccination rate was low; only 4.2 vaccine doses per 100 people were administrated by March 20, 2021. However, by October 23, 2021, this rate reached 84.5 doses per 100 people, 40.7% of the population was fully vaccinated, and 54.1% had received at least one dose.

An integral comprehension of this epidemic, from different perspectives, is needed to prevent and control it. Presently, enormous research has been accomplished, principally in the field of medicine. However, it is essential to improve our understanding of how COVID-19 spreads over a territory. For instance, the description of the epidemic characteristics, including spatio-temporal distribution and association with other features, may be useful for identifying covariates, modelling epidemic behaviour, and providing reliable information for decision-making (Franch-Pardo et al., 2020). Franch-Pardo et al. (2021) reviewed 221 scientific articles related to spatial science and COVID-19 and observed that researchers increasingly employed spatial analysis to study the impacts of the pandemic. The studies concerned different aspects of the pandemic such as spatial correlation and autocorrelation of cases number, mapping of risks and vulnerability, assessment of the impacts of preventative measures and knowledge dissemination. Nevertheless, there are few studies concerning the spatio-temporal dynamics of COVID-19 in Mexico (However, see Ghilardi et al., 2020; Hernández-Flores et al., 2020; Méndez-Arriaga, 2020; Núñez Medina, 2021; Santana Juárez et al., 2020; Tello-Leal & Macías-Hernández, 2021; Villerías Salinas, Nochebuena & Flores, 2020).

This study aims at investigating the spatial distribution patterns and dynamics of the COVID-19 in Mexico over a period of 19 months from Feb 23, 2020 (beginning of the epidemic) to October 2, 2021 by carrying out spatial analysis techniques such as the computing of spatial autocorrelation and the identification of clustering patterns of the confirmed cases over time.

We analyzed the spatio-temporal patterns of the epidemic to test the following hypothesis: COVID-19 pandemic in Mexico is characterized by different clusters evolving in space and time as parallel epidemics.

Connectivity based on a gravitational model allows explaining inter-municipality contagion better than other distances such as Euclidean, least-cost, and resistance distances.

Materials and Methods

Data sources

The data sets we used for this study are composed of the epidemiological and geographical auxiliary data: The daily number of confirmed COVID-19 cases and deaths for the period from February 23, 2020 to October 2, 2021, reported by the Secretary of Health were obtained from the Mexican federal government open data platform (https://www.gob.mx/salud/documentos/datos-abiertos-152127, accessed October 16, 2021).

Country-based COVID-19 statistics (Number of administered doses and number of fully vaccinated people) from https://ourworldindata.org (University of Oxford).

Digital maps of municipality boundaries, human settlements, and road networks from the National Institute of Geography and Statistics (INEGI), http://en.www.inegi.org.mx/datos/.

2020 Census of Population and Housing (https://censo2020.mx/) (INEGI, 2021).

The records from the Secretary of Health contain additional information such as age, sex, foreign and immigrant status and co-morbidities. Concerning information useful to spatialize the data, the records are based on the two official administrative units most commonly used in Mexico, which is a federal republic: (1) the states (31 states and Mexico City, an autonomous entity) and the municipalities (2,465 municipalities). Each record indicates the patient’s residence state and municipality and the state (but not the municipality) of the health unit where the consultation occurred. In the present study, focused mainly on spatial patterns, we aggregated data at the municipal level using the municipality of the patient residence because this administrative unit was the less aggregated level available in the data.

Since there is a delay in case reporting and confirmation, we limited this analysis to the period between February 23, 2020 and October 2, 2021 using the available information on October 16, 2021. We also discarded records in which the state of residence was different from the state where the patient did the consultation because these records correspond likely to patients who get infected outside of their municipality of residence. These records represent about 6% of the confirmed cases.

The geographical database is based on the Lambert conformal conic projection, a conic map projection established on two standard parallels, which minimizes deviation from the unit scale within a region comprising the two standard parallels. False easting and northing ensure positive coordinate values expressed in meters. Its elaboration is presented in Mas (2021). The dataset and the scripts are available at Mendeley Data to facilitate practical reproducibility (Nüst & Pebesma, 2021). Repository name: Data_Mexico_COVID19

Data identification number: 10.17632/mc37xdzw74.1 (DOI)

Direct URL to data: https://data.mendeley.com/datasets/mc37xdzw74/1

Methods

The daily information from the Secretary of Health was aggregated to the weekly level to simplify the data and avoid the day of the week bias. This temporal aggregation allows to reduce noise in the data but at the expense of loosing temporal resolution and a shorter time series (Alarcon Falconi et al., 2020). The number of new confirmed cases and the number of deaths were computed for each municipality per week during the period between February 23, 2020 and October 2, 2021.

Spatial autocorrelation analysis enables users to test whether the observed value of a nominal or ordinal variable in one place is independent of values of the same variable in neighbouring areas (Sokal & Oden, 1978). In this case, the spatial autocorrelation assessment enabled us to evaluate if the municipalities with high or low infection rates tend to be spatially aggregated and form clusters. Moran’s I index is a measure of spatial autocorrelation that can be used to explore the spatial distribution of diseases (Lawson, 2006). It has been used in various studies concerning COVID-19 (Sotela Barrantes & Solano Mayorga, 2020; Cordes & Castro, 2020; Kang et al., 2020; Yang et al., 2020). To depict the spatial association of COVID-19 cases over time, Moran’s I statistic was computed for each week (Eq. 1). Moran’s I index varies between −1 and +1. A value of -1 indicates a perfect clustering of distinct values (perfect dispersion), zero means no autocorrelation (random spatial distribution) and +1 indicates a perfect clustering of similar values.

(1) It=nS0∑i=1n∑j=1nwij(xit−x¯t)(xjt−x¯t)∑i=1n(xit−x¯t)2,

where It is the value of the Moran’s I statistic at time t, xit the number of cases at location (municipality) i and time (week) t, x¯t the average of the number of cases in the entire region and time t, n the number of municipalities, wij the weight between observation i and j, and S0=∑i=1n∑j=1nwij. In its simplest form, the weights take values 1 for close neighbours, and 0 otherwise. We also set wii = 0 because a region cannot be adjacent to itself.

During an epidemic, it is crucial to implement spatio-temporal surveillance that can prioritize locations for specific interventions, rapid tests, and resource allocation. One such method is space-time exploration statistics (Kulldorff, 1997), which is widely used to detect clusters of different types of diseases (Coleman et al., 2009; Zheng et al., 2014) including COVID-19 (Andersen et al., 2021; Ballesteros et al., 2020; Desjardins, Hohl & Delmelle, 2020; Greene et al., 2021; Hohl et al., 2020; Rosillo et al., 2021). A cluster is an unusual aggregation of disease cases grouped together in space and time (Lawson, 2018). As the following step, spatial scan statistic was applied to detect and evaluate disease clusters of newly confirmed cases using the “flexibly shaped spatial scan” approach proposed by Tango & Takahashi (2005). This algorithm can detect irregularly shaped clusters such as those along with a linear feature as a road, while algorithms based on a circular window have difficulty in accurately detecting non-circular clusters and tend to define a more extensive cluster than the true one by incorporating surrounding regions (Tango & Takahashi, 2005; Tango & Takahashi, 2012). In epidemic monitoring, the size of a cluster cannot be known a priori, and the population at risk is not evenly distributed. For instance, under the null hypothesis of equal risk of disease inside and outside the cluster, we expect more cases in an urban area compared to a rural area of similar size, due to the higher urban population density. No analytical solutions have been found to obtain the probabilities in these tricky conditions and the algorithm uses Monte Carlo hypothesis test to get the p-values (Kulldorff, 1999).

For each region (e.g. municipality), the algorithm creates a set of irregularly shaped candidate clusters aggregating a growing number of connected regions. In total, a very large number of different but overlapping arbitrarily shaped windows are created. For each candidate cluster, the number of observed cases was compared to the number of expected cases of COVID-19. We assumed that COVID-19 cases were Poisson distributed, as has been done in previous spatial COVID-19 studies using spatial scan statistic (Andersen et al., 2021; Ballesteros et al., 2020; Desjardins, Hohl & Delmelle, 2020; Greene et al., 2021; Hohl et al., 2020; Rosillo et al., 2021). The null hypothesis is that the incidence of COVID-19 is randomly distributed over space, and the alternative hypothesis is that the incidence increases inside the cluster. In order to test whether the clusters are statistically significant, the log likelihood ratio (LLR) was estimated by Monte Carlo randomization with 999 replications. The p-value was estimated by comparing the rank of the likelihood ratios from the real data set with the likelihood ratio values from the randomized data sets. If this rank is R, then p-value = R/(1 + Ns), where Ns is the number of simulations. The non overlapping statistically significant clusters were retained (p ≤ 0.05). Clusters were estimated for each week, allowing to analyze their temporal evolution by computing the date of their first occurrence and their duration over time. For each cluster, the relative risk (RR) was computed (Eq. 2). RR is the risk inside a cluster divided by the risk outside the cluster.

At the municipality level, we also calculated the Spearman correlation between population density, the date of the beginning of belonging to a cluster, the duration in a cluster, and the maximum relative risk.

(2) RR=c/e(C−c)/(C−e),

where RR is the relative risk, c is the total number of cases in a cluster, e is the total number of expected cases in a cluster, and C is the total number of cases in the country.

Finally, an attempt was made to determine which type of distance measure is better suited to explain disease spreading. For that, the distance between municipalities was calculated using different approaches: (1) Euclidean distance, (2) least-cost distance, (3) resistance distance, and (4) gravity model interaction.

Euclidean distance corresponds to a straight line between two locations based only on their coordinates (Fig. 1A). The other distances, least-cost distance, and resistance distance are based on graph theory. Graphs are obtained by connecting each cell center with its nearest neighbours, which become the nodes of the graph. Weights are associated with each edge and express the conductance (inverse of the resistance). In the present study, the road network map was rasterized using the speed limit as the value of conductance and a spatial resolution of one kilometer. Cells were connected with their eight orthogonal and diagonal nearest neighbours (Moore neighbourhood). The least-cost distance is the least costly path to travel from one point to another, taking into account the cost associated with moving through space. The distance is expressed in cost units, in this case, time (minutes) (Fig. 1B). The resistance distance allows the incorporation of multiple pathways, for instance, the least cost route and alternative ones using secondary roads (Fig. 1C).

Figure 1 Distances between two locations (blue circles).

(A) The Euclidean distance corresponds to the straight line between two places and does not consider terrain characteristics as road network and obstacles, (B) the least-cost distance takes into account the terrain conductance associated, in this case,with travelling time to determine the optimal (least costly) pathway, (C) the resistance distance integrates multiple pathways using the conductance.

Mobility and transport systems, such as trade and commuting patterns, greatly affect the magnitude and the speed of disease transmission at regional, national, and global scales (Francetic & Munford, 2021; Mitze & Kosfeld, 2021; Moghadas et al., 2020). The gravity model enables geographers to model the amount of interaction between two places (Flowerdew & Aitkin, 1982). The expected interaction between two places is proportional to the size of their populations Pi and Pj, and inversely proportional to the square of the distance between them (Eq. 3):

(3) Iij=kPiPjdij2,

where Iij is the interaction between places i and j, k is a constant depending on the specific data set, and Pi, and Pj are the population sizes of places i and j and, dij the distance between these two places. In this study, we used the Euclidean distance and we set k = 1 because we want to obtain a relative value of the interaction between locations without adjusting the model to interaction data such as the number of persons commuting between the two localities.

To determine the geographical position of each region, instead of using the coordinates of the municipality centroid, the coordinates of the municipality capital cities were used. This decision was based on the fact that, most of the confirmed cases were found in largest cities (Villerías Salinas, Nochebuena & Flores, 2020), and the distance between capital cities represented the connection between municipalities better than the distance between centroids. Additionally, it is worth noting that the gravitational model enabled us to enhance the relationship between the largest cities because it considers the size of the population.

For each week, the municipalities already belonging to a cluster in the previous week (time t−1) are identified (blue region in Fig. 2). We calculated the distances between each one of these municipalities and the other municipalities, including the newly incorporated into a cluster (orange regions in Fig. 2) and those which remained outside of any cluster (white regions in Fig. 2). The distances were computed using the four approaches described previously. The type of distance that explains the spread of the epidemic better is expected to express a more significant closeness between the pre-existing clusters and the newly incorporated municipalities. Therefore, considering the municipalities infected at week t−1 as the origin of subsequent contagions, we can consider the distances as possible predictors of the incorporation, or not, of the municipalities into a cluster at the following week t.

Figure 2 Paths between infected municipalities at time t−1 and infected and non-infected municipalities at time t.

The length of the shortest path between two points is the Euclidean distance. Other types of distance will be calculated using the same sets of municipalities.

We carried out a Receiver Operating Characteristic (ROC) analysis (Mas, 2018; Metz, 1978) to assess the distance’s performance in classifying the municipalities in two categories (inside a cluster or not). We calculated a ROC curve from the independent risk score expressed by the inverse of the inter-municipality distance merged to data containing observed binary outcome variable (weekly infected and non-infected). The risk score values were sorted in ascending order and used as successive cutpoints. The continuous risk variable was dichotomized for each cutpoint, producing a classification of the observed event: non-infected for the lower risk class corresponding to the larger distances and infected for the higher risk/lower distances class. Then, this binary classification was compared to the observed outcomes, and false-positive and true-positive rates were computed. The false-positive rate is the proportion of municipalities classified as infected but that do not belong to a cluster. The true-positive rate is the proportion of municipalities classified as infected and belong to a cluster. Plotting the false-positive rate (horizontal axis) and the true-positive rate (vertical axis) over the range of possible cutpoints produces the ROC curve. The area under the ROC curve (AUC) represents the degree of the predictor’s separability and varies between 0 and 1. A value of AUC of approximately 0.5 indicates that the model has no discrimination capacity (random model). A value close to 1 indicates a model with high discrimination performance. AUC below 0.5 indicates that the model has less predictive power than a random model (Fawcett, 2006; Mas et al., 2013; Metz, 1978). We performed ROC analysis using the n shortest distances from the municipalities infected at week t−1 (n nearest neighbours) with different values of the number of neighbours considered (n = 5, 20, 50, 100, 250 and 500).

All the analyses were carried out using the R program (version 4.0.2) (Core R Team, 2020), in particular the packages FlexScan 0.2.0 (Tango & Takahashi, 2012), gdistance 1.3-1 (van Etten, 2017), rflexscan 0.3.1 (Otani & Takahashi, 2020), reportROC 3.5 (Du & Hao, 2020), sf 0.9-6 (Pebesma, 2018), and spdep 1.1-5 (Bivand, Pebesma & Gómez-Rubio, 2013).

Results

Figure 3 shows the weekly number of COVID-19 confirmed cases and deaths in Mexico during the period between February 23, 2020 and October 2, 2021. The curve exibits three epidemic waves or phases with a rising number of confirmed cases, a peak, and then a decline. The number of confirmed cases reached the first peak with 50,000 per week during week 29 (July 12–18, 2020), decreases to less than 30,000 weekly cases (week 38) and increases again to reach the second peak with more than 100,000 new confirmed cases by week (weeks 54–56, January 3–23, 2021) and the third peak with more than 130,000 newly confirmed cases (week 85, August 8–14, 2021). The three consecutive peaks present increasingly steeper slopes, indicating higher rates of infection over time.

Figure 3 Number of weekly confirmed COVID-19 cases and deaths in Mexico between February 23, 2020 and October 2, 2021 (weeks 9 to 92).

The first and third waves correspond to peaks of the number of clusters and the number of municipalities included in a cluster (Fig. 4A). The incorporation of new municipalities into clusters and the re-incorporation of other ones that previously belonged to a cluster precede these two peaks of confirmed cases. The second wave has a different pattern and concerns mainly municipalities included into a cluster previously (Fig. 4B). As shown in Fig. 4C, the first few clusters (weeks 10 to 15) present a high relative risk because there were almost no reported cases outside the clusters. The clusters’ relative risk is variable, as depicted by the grey shaded area between the 10 and 90% quantiles. As the number of cases increases to reach a peak, the relative risk decreases. As the epidemic develops, the proportion of cases inside a cluster tends to increase (Fig. 4D). This proportion fluctuates between 45% and 73% of the weekly newly confirmed cases. In the cases of the second and third waves, the steep increase in the number of cases is accompanied by a rapid decline in the proportion of cases belonging to a cluster. By October 2, 2021, 97.4% of the municipalities have at least one confirmed case.

Figure 4 Clusters characteristics between February 23, 2020, to October 2, 2021 (weeks 9 to 92).

(A) Number of clusters and number of municipalities included into clusters, (B) number of municipalities incorporated for the first time into a cluster and number of municipalities re-incorporated into a cluster, (C) median relative risk value of the clusters. The grey-shaded region represents the area between the 10% and 90% quantiles, (D) proportion of the new confirmed cases belonging to a cluster.

Moran’s I index expresses the correlation of the cases’ number between neighbouring municipalities. Its value fluctuates between 0.15 and 0.5, except for weeks 9 and 10 which have a low number of cases. These Moran’s I index values indicate a significant spatial dependency of the number of newly confirmed cases and deaths. The peak of the first wave corresponds to a low value of the Moran’s I index. In the case of the second and third waves, the peaks of confirmed cases coincide with peak values in the index curve. The Moran’s I indices based on the number of new cases and the number of deaths show similar curves. However, the index based on the number of fatalities presents lower values and an offset of approximately 2 weeks (Fig. 5).

Figure 5 Moran’s I index for the number of new weekly cases and deaths (p-values < 0.05).

The “flexibly shaped spatial scan” was applied to weekly confirmed cases to detect clusters’ formation. Figure 6 presents the detected clusters and their relative risk at four key dates distributed over the study period: Week 13 corresponds to the beginning of the community transmission, and weeks 29, 55 and 85 coincide with the peaks of the first, second and third waves of the epidemic, respectively (see Fig. 3). We can observe that, since the first weeks of the pandemic, clusters are present in very distant places of the territory.

Figure 6 Clusters and relative risk at different dates.

(A) Week 13, (B) Week 29, (C) Week 55, (D) Week 85. The inset map represents Mexico City (blue boundary) and its surrounding area.

Figures 7 and 8 allow discerning some patterns of contagion, where more recent clusters surround early ones. These patterns of neighbouring effect are consistent with the positive value of the Moran’s I index previously founded. As shown in Fig. 9, some municipalities, often the more densely populated as Mexico city, remain a considerably long period in a cluster (see the map of municipalities population density in the Supplemental Materials). It is also worth noting that 36.5% of the municipalities were included in a cluster at some moment during the period of study (See the complete series of weekly clusters’ maps in the Supplemental Material). The correlation analysis indicates that municipalities with higher population density tend to be integrated into a cluster earlier and for a longer time but present a lower relative risk (Fig. 10).

Figure 7 Patterns of contagion in Mexico City (blue boundary) and its surrounding area during weeks 9 to 17.

The cluster includes municipalities in contact with previously infected ones successively over time.

Figure 8 Date of clusters’ formation.

The inset map represents Mexico City (blue boundary) and its surrounding area.

Figure 9 Clusters’ duration (number of weeks).

The inset map represents Mexico City (blue boundary) and its surrounding area.

Figure 10 Correlation between the population density, the number of the week of the first inclusion into a cluster, the cluster duration and the maximum and median relative risk.

Values indicate the Spearman correlation. In the scatter plots, a logarithm transformation was applied to population density and relative risk.

The neighbouring relationship between municipalities vary significantly depending on the type of distance used to define proximity. For instance, Fig. 11 shows the 20 nearest neighbours of Guanajuato’s municipality based on the four distances. The Euclidian distance selected the neighbours into a circle around Guanajuato. Least cost and resistance distances which consider road networks lead to selecting different neighbouring municipalities in a slightly another way. Based on these three distances, the neighbouring municipalities are continuous in space. The gravitational model gave a more contrasting result. Because it takes into account the size of the municipalities’ population, it integrates neighbouring municipalities with large cities (Aguascalientes, Lagos de Moreno and San Luis Potosí in the North, Morelia in the South, and Guadalajara in the West). Therefore, the 20 nearest neighbours are part of a network linking the more populated municipalities at a regional scale. The neighbouring obtained using the different types of distances and a different number of neighbours reflects various spatial relationships at a different scale (see neighbours obtained with different numbers of municipalities in the Supplemental Material).

Figure 11 Twenty nearest neighbours (blue polygons) of Guanajuato (orange polygon) based on the different types of distance.

(A) Euclidean distance, (B) resistance distance, (C) least cost distance, (D) gravitational model.

The AUC obtained by the gravitational model is higher than the one obtained by the other distance metrics. Therefore, the connectivity based on this model enables us to explain inter-municipality contagion from a spatial perspective better than the other distances used in the comparison as Euclidean, least-cost, and resistance distances (Fig. 12). The gravitational model’s better performance is accentuated when the number of neighbours increases (see figures obtained with n = 5, 10, 50, 100, 250 and 500 neighbours in Supplemental Material), that is, when we observe the relationship between municipalities at a coarser scale. This result suggests that the contagion pattern depends, as expected, on the strength of the interaction between places. The number of persons commuting between these places is related to this interaction level and likely to contagion. When statistics on the number of commuters is lacking, a straightforward model like the gravitational one can give a realistic picture of the strength of interaction between cities or municipalities. However, the AUC is relatively low (average value of 0.6), indicating an insufficient prediction power and shows considerable variation over time (Fig. 13).

Figure 12 AUC obtained using different types of distances to predict weekly newly infected municipalities using the 20 nearest neighbours over the period February 23, 2020–October 2, 2021.

Average AUC values are 0.49, 0.60, 0.49 and 0.42 for Euclidean. gravitational, least-cost and resistance distance, respectively.

Figure 13 AUCs obtained weekly using the 20 nearest neighbours and different types of distances to predict weekly newly infected municipalities.

Discussion

Moran’s I index positive values are similar to values observed by Kang et al. (2020) in China and indicate that the number of newly confirmed cases and deaths show significant spatial dependency (Fig. 5). Moran’s I index fluctuations correspond to different spatial patterns of the spreading of the epidemic, from clustered to more random patterns, where nearest neighbouring municipalities tend to present similar or contrasting numbers of cases, respectively (see maps showing the numbers of weekly new cases in the Supplemental Materials). We also observed that, since the beginning of the pandemic, clusters are present in very distant places of the Mexican territory. As the epidemic began in Mexico 1 month after Europe, at the early epidemic stage, the virus was introduced in different areas by people coming back after travelling abroad. The first COVID-19 cases showed up in states located at the north (Coahuila, Durango), centre (Mexico City, Puebla, State of Mexico) and south-east of the country (Chiapas, Tabasco). The different clusters evolve in space and time as parallel epidemics. De Anda-Jáuregui (2020) found similar results using an information-theoretic measure of statistical dependence to reconstruct a network connecting municipalities with similar daily new cases behaviours. He identified several modules with high within-connectivity, which constitute coexisting epidemics regardless of geographical distance.

The space-time scan statistic enables us to detect clusters at different dates during the development of the epidemic. The flexibly shaped version was helpful because many of the clusters present an elongated shape. These results may help obtain a rapid picture of the COVID-19 hotspots and provide insights to design effective control and preventive strategies of social distancing policies, testing or vaccination. In particular, updating the confirmed cases data, the space-time scan statistic allows a daily tracking of emerging clusters present at the end of the study period by evaluating their size, relative risk and location.

However, the efficiency of such tracking will depend on the rapidity of confirmed cases reporting. For instance, only 57% of the confirmed cases, which hospital admission date is Jul 8, 2020, were confirmed and integrated into the database on Jul 13, 2020. The database incorporated 90% of these cases on July 31, 2020. This lag of time between patient admission and information incorporation into the database is an obstacle in using space-time scan statistic to daily track emerging clusters, particularly because the delays are likely unevenly distributed over the territory. Using municipalities, which are relatively large administrative units, to aggregate the data is another limitation of this study. While aggregated spatial data at the municipal level have utility at the national political scope, they offer little efficacy to contain the pandemic locally (Franch-Pardo et al., 2021). Anonymous detailed data sets allow the design of specific management strategies with greater possibilities of breaking the chains of infection. Combining high-resolution data with fieldwork further increases the probability of significantly reducing spread (Ghilardi et al., 2020). However, the availability of information at high spatial and temporal resolution presents issues related to privacy. Additionally, it is worth noting that space-time scan statistic can also be applied to other important diseases in Mexico as chikungunya, dengue and zika (Bisanzio et al., 2018).

Another limitation concerns the completeness and representativeness of the cases reported by the database. A seroepidemiological survey based on a robust sampling (i.e. avoiding small or non-random selection of participants) can provide more precise estimates of seroprevalence in the population than analysis based on confirmed cases (Pollán et al., 2020). However, Mexico testing policy focuses on persons with COVID-19 symptoms and key groups (Hale et al., 2021). The virus’s testing level throughout the entire period was low: Between February 23, 2020 and October 2, 2021, the average Daily COVID-19 tests per 1,000 people varied between 0.1 and 0.4. Due to the existence of many asymptomatic cases and the reduced number of diagnostic tests, the epidemiological surveillance of confirmed cases represents only a proportion of all infections (Pullano et al., 2021; Wu et al., 2020). Therefore, this information’s limitations are that (a) confirmatory testing is strongly biased towards symptomatic patients, (b) an increase in testing can lead to a rise in reported confirmed cases, and each state makes its own data collection. However, we do not expect large variations in the number of cases due to an increase or decrease of the testing efforts because testing was low during the entire period, without huge variations. Since the federal government (Secretary of Health) mediates the data aggregation, some methodological consistency allows for homogeneity and reasonable comparisons between states and municipalities. Moreover, the Health Secretary’s confirmed cases dataset was the only available source at the national level.

The spreading of the pandemic over the Mexican territory is complex, with phases dominated by clusters that alternate with periods when the number of new cases is more evenly distributed. At the municipality level, we found that the population density was positively correlated with an earlier inclusion and the duration inside a cluster but negatively with the maximum relative risk. However, the correlation coefficient values were relatively low (absolute values between 0.24 and 0.31 see Fig. 10). Several studies found similar relationships of outbreak characteristics and population density (Diao et al., 2021; Sy, White & Nichols, 2021; Wong & Li, 2020). These authors found that many other variables such as social distancing and strictness of lockdown, climatic variables and poverty play also an important role in the epidemic spreading. High population density increases the contact rate of an individual, and therefore the rates of transmission, leading to more rapid and larger outbreaks. However, larger cities are more connected (e.g. airports), so it seems logical that they were first infected and present a larger number of cases than smaller and more isolated cities. It is worth noting that municipalities with higher population densities are slightly related to lower relative risk. According to Jo, Hong & Sung (2021), connectivity rather than population density plays a more important role in the spread of infectious diseases such as COVID-19.

Another factor that can considerably affect the pandemic dynamics is vaccination (Moghadas et al., 2020). In Mexico, the vaccination campaign began at the end of 2020, but the vaccination rate was only 1% of the population (fully vaccinated) on April 6, 2021. However, since this date, this rate has increased regularly and reached 35.7% fully vaccinated and 50% with at least one dose at the end of the studied period (October 2, 2021). Therefore, we can expect an effect of the vaccination on the spread of the disease, at least at the end of the period. However, available data only concern the total number of administered doses and fully vaccinated persons. There is no information aggregated at the state or municipality level, which could enable an analysis of the effect of vaccination on the spatial distribution of confirmed cases.

An important topic when using aggregated data is the potential effect of aggregation on statistical analysis. This effect, known as the Modifiable Area Unit Problem (MAUP), is a form of ecological fallacy (Openshaw, 1984). It has rarely been evoked in the studies about the Spatial patterns of the COVID-19 epidemic, except by Wang & Di (2020), who showed that MAUP could influence the relationship between COVID-19 and atmospheric NO2. In the present study, the effect of MAUP is limited because we did not intend to establish relationships between variables (e.g. incidence rate versus factor risks). However, using the total municipality population represents a simplification that does not take into account the distribution of the population inside the municipality. The contagion patterns are likely very different between municipalities in which most of the population is concentrated in the capital city and others where the population is distributed among many small settlements (Garland et al., 2020). Future research will focus on the effect of spatial aggregation on the relationship between the lethality rate and risk factor related to COVID-19.

An important issue is to develop models able to make accurate predictions of future hotspots of the disease. We found that the strength of interaction between municipalities was a predictor of contagion. However, as pointed by Akhmetzhanov et al. (2020), publicly available data on public and private transport travel are often inconsistent and scarce. The mathematical models of human movement are a promising alternative to replacing observed transportation data. The gravitational model’s advantage is its simplicity. The connectivity between two places is based only on the population and the distance between the two places. Nevertheless, the power of prediction of the model, assessed by the AUC, is relatively low (AUC = 0.6) and shows considerable variation over time (Fig. 13). Spatio-temporal modelling of infectious disease transmission and outbreaks is highly challenging because of its complexity in intertwining biological systems and social systems (Dong et al., 2021). In future research, additional variables, such as population density and socio-economic conditions, should be considered to improve the model.

Conclusions

We applied a geostatistical analysis to detect spatio-temporal patterns of the COVID-19 epidemic at the municipality level over 12 months. We found that the COVID-19 pandemic in Mexico is characterized by different clusters evolving in space and time as parallel epidemics. These clusters likely originated in very distant locations and varied conditions due to environmental and social diversity. Connectivity estimates based on a gravitational model explain the contagion pattern between municipalities better than other distances commonly used in geography such as Euclidean, least-cost, and resistance distances. These connectivity measures can be used in models aimed at predicting epidemic spreading over a territory and designing strategies to contain the spread of the COVID-19 pandemic.

The present study results may help map the COVID-19 hotspots and design effective control and preventive strategies. In particular, updating the daily confirmed cases data, the space-time scan statistic allows tracking emerging clusters.

Supplemental Information

Supplemental Information 1 Weekly maps of COVID-19 distribution.

Maps of weekly new cases number and clusters with associated relative risk

Click here for additional data file.

Additional Information and Declarations

Competing Interests

Author Contributions

Data Availability

The authors declare that they have no competing interests.

Jean-François Mas conceived and designed the experiments, performed the experiments, analyzed the data, prepared figures and/or tables, authored or reviewed drafts of the paper, and approved the final draft.

Azucena Pérez-Vega analyzed the data, prepared figures and/or tables, authored or reviewed drafts of the paper, and approved the final draft.

The following information was supplied regarding data availability:

The data is available at Mendeley: Mas, Jean-François (2020), “Data_Mexico_COVID19”, Mendeley Data, V1, doi: 10.17632/mc37xdzw74.1.

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
