# Peer review of "Spatiotemporal patterns of the COVID-19 epidemic in Mexico at the municipality level"

_PeerJ, doi:10.7717/peerj.12685_

## Round 0.1 · original submission · Major Revisions

Experts have reviewed your research and found it requires major revisions before it can be suitable for publication. Please respond line by line tracking all your changes and provide rebuttal where you see no changes or edits are needed.

Reviewer 1 ·

Basic reporting

The papers applied a few spatial statistics to the COVID-19 data in the country.

Experimental design

fine

Validity of the findings

too simple

Additional comments

Suggestions:
1. Figure 4. Calculate and add the spatial stratified heterogeneity (www.geodetector.cn)
2. Compare and interpret the statistical findings in epidemiology in deep.

Reviewer 2 ·

Basic reporting

The work is interesting, but it lacks many points, detailed below, for being able to be accepted. Several references need to be added, more suitable technical words be used and clarity is needed in several parts.

Experimental design

The question is well-defined and the methods are explained in detail.

Validity of the findings

I am missing a longer discussion, taking into account that the COVID-19 literature is likely to be used for policy-making. Then, I suggest the author to contextualize his results and discuss more broadly their consequences.

Additional comments

Fig. 1: this figure does not show the distances, it only displays some straight lines between the centroids of the geographical units. I suggest adding distances using a color code. I think that all the distances used in the work need to be included in this figure, illustrating the differences.
The word contaminated, which appears several times throughout the manuscript, is not the technical work used in the context of epidemics.
L19: 'at this date' is ambiguous, as the article may be read at any time, maybe 2050. I suggest saying 'By December 1, 2020'.
L21: 'and,'->'and'
L41: can you include some references?
L60-64: can you provide the digital sources of these datasets?
L71: references needed, I suggest to cite Pullano et al. Nature 590, 134-139 (2020), and Eguíluz et al., Frontiers in Medicine 7, 563455 (2020).
L78-79: not clear. 'Low levels of testing' means low positivity rates or low testing effort. Please clarify and, if possible, quantify this sentence. I do not agree with the statement saying that the number of cases will not change if testing effort is modified, taking into account the high number of asymptomatic cases.
L101-104: please reference Fig. 3 here.
Eq. (1): if I understood correctly, I depends on time, as it is computed weekly, but there is not time-dependence in this equation. Please specify it.
Question: what defines a cluster? Are the authors trying with random combinations and random sizes and keeping only those statistically relevant? These is an effort to introduce this in the manuscript, but more clarity is needed.
Fig. 4: I think that, for understanding better the spreading, this figure needs several panels, additionally to that already included: 1) temporal evolution of the number of clusters, 2) temporal evolution of the cluster geographical coverage (individual, average, or median).
L243: I think that 'amount' should be replaced by 'strength'
As I mentioned above, the final section needs more background and discussion.

Reviewer 3 ·

Basic reporting

I believe this manuscript has potential, but requires additional work:

Authors should note more explicitly the assumptions in their analysis, e.g. lines 65-68 they are essentially assuming age and co-morbidity are distributed uniformly across municipalities (since it is excluded from the analysis) and lines 82-85 assuming no imported cases. They could then revisit this in comparison with the comment from line 137. For this reason, I also don't think the authors can conclude lines 250-252 since those effects were aggregated out of the model.

It is unclear to me what the purpose of the comparison mentioned in lines 101-104 adds since the analysis is done for Mexico only and the hypotheses presented by the authors do not mention comparison with other nations. For this reason I would omit Figure 3 and lines 206-213. The comparison in instead with Kang et al. 2020 and might be good to mention that this is work analysing cases in China.

Experimental design

Authors should specify timing to allow for increased options for reproducibility. Some of the current "dates" are not informative/defined
line 24 (end of March)
line 109 (end of March)
line 201 (beginning of the epidemic)
lines 203-204 (end of November)
line 226 (beginning of the epidemic)
Additionally "as the epidemic began in Mexico one month after Europe" is very vague.

Authors should add the time mentioned in their expression for Moran's I index. They mention is it time-dependent and so should have something like I_t, x_{it} and \bar{x_t} where $t$ denotes week and \bar{x_t} = \frac{1}{n}\sum\limits_{k = 1}^n x_{kt}.

Authors should note what the constant k represents (equation 3)

I would also add expressions for/calculations of D_c and D_{nc} by d_{ij} from (3)

What is the inset in figures 5, 6, and 7?

Authors should note what a good value for per cent relative difference is (Figure 8)? Is this a large value in absolute terms (indicating the algorithm's ability to detect clusters? Authors should explain this in more detail, including why some are negative.

Validity of the findings

The first peak mentioned on line 218 does not correspond with what is shown in Figure 4 where the first peak seems to occur around week 10

Line 254: My understanding is that scan statistics do not seek to prevent epidemic spread but rather detect anomalies (hence outbreaks/clusters) and so I think the wording is a bit too strong here

The authors might want to mention (line 221) that deaths might be a more "true" measure since all COVID-19-related deaths can be expected to be reported (since severe cases will lead to deaths) and so the authors could revisit their brief mention of the effect of underreporting mentioned earlier in the manuscript

The authors might want to note that the use of weekly case counts allows them to disregard day of the week effects (or they are subsumed), but that this is at the expense of having less information/a shorter time series.

Should extend final sentence (line 184) further.


The phrasing of the sentence in line 238 is a bit odd.

Additional comments

The colours mentioned in the text (lines 186-194) do not correspond with those used in Figure 1

I currently do not have access to a printer so I have not been able to check if all of the figures print well in black and white and would encourage the authors to do so.

Line 19 - I would suggest the authors add the year as there is a chance COVID-19 might still be around in Dec 2021
* * *
Additional literature the authors may want to consider:

Akhmetzhanov et al. https://doi.org/10.1101/2020.04.24.20077800 includes some similar expressions to equation 3 and I believe use the same package in their calculations

Xia et al. https://doi.org/10.1086/422341 for use of a gravity model in an infectious disease application may provide insights into expanding the sentence on lines 183-184

Considerations regarding popultion density: https://www.scienceinparliament.org.uk/wp-content/uploads/2020/08/johnson_PSC.pdf
* * *
Miscellaneous minor corrections:
* line 134 should be "tend" as "algorithms" is plural
* line 218"COVOID-19" should be "COVID-19"
*When defining calculations, authors should not note . Remove "RR is the relative risk" (158) and move "RD is the relative difference between the two sets to distances" (line 195) to before the equation.
* Inconsistencies in terminology in paragraph starting on line 214 ("Morans's I statistic", "Moran index", and "Moran") and again in line 236 ("Moran statistic")
* Suggest authors add "in 2020" to caption for Figure 2
* Remove apostrophe in caption for Figure 6
* Add space after apostrophe in caption for Figure 7
* I am not sure the long/lat axes are needed for figures 6 and 7
* Inconsistencies in figure captions wrt whether a full stop is included at the end or not
* Inconsistencies in in-line mathematical symbols, would suggest authors go through the text and ensure everything is in italics to match equations (e.g. lines 158-159, 196-198, caption Figure 1, 188, but I have missed some).
* The scale and compass marker could be removed from figures 5, 6, and 7 as they do not add much
* Suggest adding "from a spatial perspective" after "contagion" on line 243

---

## Round 0.2 · Major Revisions

The manuscript received mixed reviews, and while several noted the potential of your work, at least 1 reviewer expressed frustration that their comments were not completely addressed and all noted several areas where justification is needed, figures improved and grammatical and language changes due. Furthermore, formulas 1 to 4 have undergone quite a bit of changes between the manuscript submissions, they should be referenced at this point specially that they arent derived but are rather previous work. Hence, it is my opinion that the manuscript is not ready for publication.

Please respond to the reviewers' new comments, make the changes suggested or provide a rebuttal why you do not consider all or some of the suggestions suitable.

I look forward to your revised manuscript and response to the reviewers' comments.

Best wishes,
Editor

Reviewer 1 ·

Basic reporting

please see my last report

Experimental design

please see my last report

Validity of the findings

please see my last report

Additional comments

1. The paper applied Moran’s I and Satscan to illustrate hotspots of COVID19 in Mexico. It is still too simple to be an academic paper and little interesting.
2. Considering a highly diverse population, measure and attribute the spatial stratified heterogeneity (SSH) of the diseases and their change in time. Interpret the statistical findings in epidemiology.
3. My concerns are not appropriately addressed.

Reviewer 2 ·

Basic reporting

I did not like this revised version, basically because the authors try to avoid addressing some of the points raised by the reviewers, and even in one case they say they did something which in fact does not appear in the submitted manuscript.
First of all, the authors should revise the basic geometry concepts of path and distance for addressing my comments about Figure 2. A path is a line, while a distance can be calculated using that line, but it is a number. Also, in that figure, the color code for the lines ('distances') is not explained.
Authors mention that they have cited the suggested works but they did not, or at least these references do not appear in the manuscript that I received.
When authors mention that they applied spatial scan statistic, they do not mention to what they applied this algorithm (to cases, deaths, distances?).
Additionally, the new figure 7 looks interesting and may align with my previous comment, but it would be nice to see how a specific cluster evolves in consecutive weeks, to illustrate, for example, the spatial spread phenomena.

Experimental design

No comments

Validity of the findings

No comments

Reviewer 3 ·

Basic reporting

As an additional author is now listed, authors should note their Contributor Roles Taxonomy (CRediT)

Authors should add results to the abstract (suggest placing before "This study helps us understand...").

"head city" is not an English term, suggest "largest" or "capital" city used instead.

Authors should note on line 195 that area under ROC curve abbreviates to AUC

In line 197 the authors should use the number 1 to be consistent with the rest of the text (explanation of I index etc.)

In line 200 the authors should note what n is (number of neighbours considered) to aid the reader

Experimental design

Authors dates are consistent in terms of both formatting (full versus abbreviated month names) as well as content (23rd February 2020 and 7th March 2021 vs 6th March 2021). They should ensure the correct and consistent study dates are used.

Authors may wish to use a negative binomial rather than a Poisson model, allowing for more variance.

Lines 319-320 Why haven't the authors done this? Such information should be available

Please explain in more detail how the false positive and true positive rates are calculated and enter ROC/AUC

Validity of the findings

Lines 202-211: For the reasons outlined by the authors it makes little sense to compare them; would suggest the authors retain focus on Mexico

Please split Figure 5 into two separate plots with separate y-axes and add a line where RR = 1 to show where no difference in relative risk occurs

It is a bit of a stretch to conclude the gravitational model is best with the amount of overlap seen in Figure 11

Additional comments

Please format the y-axis labels as numbers (Figure 4). Could use par(las = 1) to have them horizontally.

What determines the bin width in the histograms in Figures 8 and 9?

If noting which references are pre-prints, it seems De Anda-Jauregui should also have such a label

Reviewer 4 ·

Basic reporting

This study was well conducted and reported. However, there were several limitations in the manuscript like failure to report asymptomatic cases, focusing on only two municipalities without stating any justification for this action. There is no discussion of any prophylactic treatments and vaccinations in Mexico which gives the impression that only the Jornada Nacional de Sana Distancia campaign was being used to curb COVID-19. It would be interesting to see the author incorporate these issues and information on recovered individuals in the municipalities.

Experimental design

The research questions and hypothesis are properly addressed. The methods are also well explained.

Validity of the findings

Further discussion of the results and their impact on policymaking on COVID-19 prevention measures like lockdown and vaccination is needed. These measures in addition to prophylactic treatments have been reported to reduce on the magnitude of the COVID-19 pandemic. A study in the USA reported that with limited protection against infection, vaccination could have a substantial impact on mitigating COVID-19 outbreaks (Moghadas et al., 2020). This could help better inform the audiences.

Additional comments

Abstract
Line 12-17: the abstract is quite informative, but more details should be incorporated. It seems to be focusing mainly on the methods used in the study. I suggest stating a brief COVID-19 background, Euclidean, gravitational, least cost, resistance distances. Gravitational distance was the best model to predict new infected municipalities though the AUC was relatively low and varied over time.

Introduction
Line 20: I would suggest adding some references.
Line 26: I think a brief inclusion of the mode of transmission of COVID-19 would be helpful to future readers.
Line 27: did this include complete restriction of movement and enforcement of continued use of PPEs by the public. Were there any pharmacological treatments during this period. If not please state it.
Line 54: It is not clear why it is a bold header, but I suggest creating a “data sources” section in the methods instead.
Line 67: are there records of the municipality of the health unit where patients consulted?
Line 88: the records discarded were based on the difference in the state of residence and state where the patient did the consultation. I would imagine that since the records of municipality address are available, using this and the municipality where the patient did consultation would be better since the stay is at the municipality level.
Line 89: has this been reported before? Or it is a hypothesis. If yes, is it statistically tested?

Methods
Line 114: I assume Mar 7, 2021, is a typo?
Line 115-117: please define x̅ t in equation (1), (average of the number of cases in the entire region)
Line 125: please include the reference for this cluster definition.
Line 164: what is the rationale of using Kilometer as the metric units, is this the metric unit used in the coordinate system you chose? I think the metric units should be the same.
Line 192: please include more references on ROC analysis.

Results
Line 204: Figure 3 does not show the drastic reduction of the number of deaths but rather the deaths are constant between 50 and 200 days. Please review the figure
Line 206-211: I would suggest including this in the discussion.
Line 224: In Figures 4,5 and 12; are “abr” and “dic” typos for the months of April and December respectively?
Line 240: figure 7; why was only Mexico City include in the inset map? Could it be that Mexico City reported the highest number of cases and clusters? I would suggest including a table showing the number of cumulative COVID-19 cases population and population density of the municipalities. This will help inform an international audience on what was referred to as a densely populated municipality in this study. Please label the most densely populated municipalities, they could also reveal a unique trend.
Line 244: “It is also worth noting that 26% of the municipalities were included in a cluster at some moment during the period of study.” I suggest stating the specific weeks of the mentioned period.
Line 247: in figure 10: why did not the author stick to reporting Mexico City and reported Guanajuato’s municipality instead.
Line 260: I would suggest reporting the actual AUC values in Figure 11 description.
Line 270: please choose a better line pattern for the gravitational model. The current one makes the figure challenging to comprehend.

Discussion and Conclusion
Line 273: please specify the territory being referred to here.
Line 274: do you anticipate that most of the new COVID-19 cases were due to infected individuals returning from Europe?
Figure 8 & 9: I would suggest a further discussion of why the clusters remain for longer periods of time in the most populated cities.
Line 285: I agree with the authors, however, it would be helpful to state how robust the testing, prophylactic treatment, and vaccination are in Mexico.
Line 309: could head city be meaning capital city?
Line 320: did this affect your spatial analysis?
Additional discussion inclusions: I think testing being biased towards symptomatic cases could be a limitation in the study since asymptomatic cases could also transmit the disease. I think with the emergency and adoption of COVID-19 vaccines, commenting about it as a preventive strategy and how it could affect these spatial-temporal patterns is ideal. Analyzing spatial tendencies of the recovered cases could also help in modeling control and prevention measures of COVID-19.
Additional comments
Line 16-17: typo help not helps. Please double-check the “and” used numerous times.
Line 46: include “such”/ rephrase the sentence for better understanding.
Line 52: typo inter-municipality not intermunicipality
Line 270: figure 11 typo newly not new
Line 293: typo - distributed not distribuited

Reference
Moghadas, S. M., Vilches, T. N., Zhang, K., Wells, C. R., Shoukat, A., Singer, B. H., Meyers, L. A., Neuzil, K. M., Langley, J. M., Fitzpatrick, M. C., & Galvani, A. P. (2020). The impact of vaccination on COVID-19 outbreaks in the United States. MedRxiv : The Preprint Server for Health Sciences, 1–16. https://doi.org/10.1101/2020.11.27.20240051

Annotated reviews are not available for download in order to protect the identity of reviewers who chose to remain anonymous.

---

## Round 0.3 · Major Revisions

Experts have reviewed your revised manuscript and in agreement with them it still requires major changes before it can be deemed acceptable for publication. Please address their reviews line by line, also two reviewers noted their comments may have not been addressed so please be sure to respond either with the changes required or a rebuttal.
Thanks and I look forward to seeing your updated manuscript.
best wishes,
Sharif

Reviewer 1 ·

Basic reporting

1. My concerns are not appropriately addressed so the problems remained unsolved;
2. Compare gravity model and SEIR model in prediction of infection transmission;
3. Figures 3-6. Try to interpret the time series using behind spatial process.

Experimental design

see the above

Validity of the findings

see the above

Reviewer 3 ·

Basic reporting

See below

Experimental design

See below

Validity of the findings

See below

Additional comments

I look forward to reading the point-by-point responses to the previous comments from reviewers 1 and 2 in the next revision. For this reason, I only provide a few comments at the current stage:

The references in the current version seem to be broken; they all render as "?".

Ad. my comment about using a negative binomial: I believe there may be some misunderstanding. Using a Poisson model is not a more "general purpose" approach, as it necessitates that the variance is the same as the mean. A negative binomial distribution would allow for more heterogeneity to be captured. It would seem that negative binomial options have been explored by the authors of the methodology used, cf. https://doi.org/10.1111/j.1541-0420.2010.01412.x so I hope the authors are able to try such an approach to see what effect that has on the results and provide their analysis code as open source.

Having multiple axes in figures remains confusing and I would suggest that less is more. Figures 4-6 and 14 have different time scales at different magnitudes. Readability would be improved if the x-axis above the chart were the main axis (i.e. place below the chart) and remove weeks. Similarly, in Figure 5 it would be easier to read the values of the two time series if they had separate y-axes as they are not on very different scales, one even seems to be a log scale. Having a Figure 5 with two side-by-side panels would be easier to decipher. While the descriptive plot in Figure 3 (it does not arise from the methods used and so should not really be in a Results section) is more readable, I would like to reiterate to the authors that unless case and death definitions and surveillance systems (testing, reporting, etc.) are the same across countries, it makes little sense to compare countries. I would once again suggest they retain focus on comparing the different options for distance metrics.

RE: Lines 37-41 There has not been much fomite transmission from contaminated surfaces for SARS-CoV-2 virus (https://doi.org/10.1016/S1473-3099(20)30561-2), please update to reflect that COVID-19 is a mostly airborne disease, see e.g. recent WHO review https://doi.org/10.12688/f1000research.52091.1

Reviewer 4 ·

Basic reporting

The authors properly report their methodology and findings.

Experimental design

Please see my last report

Validity of the findings

The author states that they mentioned the “huge effect of vaccination on mitigating the COVID-19 outbreaks”. However, this effect is not described in the manuscript.

Additional comments

Comments for the author
Abstract
The current incorporation of the methods and results by the author is very informative to an international audience.

Introduction
Line 28-30: It is not clear why you have the (March 23, 2021) date in brackets. Please rephrase and improve the grammatical flow.
Line 54: Are these global delays and shortages of COVID19 vaccines?
Line 57: Please rephrase “A large body of research”, I recommend “enormous research”.

Materials and methods
Line 88: Please maintain the “tense” throughout your reporting to remain chronological.
Line 89-91: I think you should state (but not the municipality) on its own and rephrase the sentence to give better meaning.
Line 93: What do you mean by “robust sampling” in this case?
Line 96-107: I would suggest stating the limitations after reporting your research findings. Perhaps in the discussion.
Line 115-118: I would suggest reporting this at the end of the methods section after describing how you obtained your results.
Line 123: Is “Mas (2021)” an abbreviation or a typo?
Line 129-132: Is there a justification for choosing France as a representative country for the whole of Europe?
Line 130-132: Please state the figure being referred to.
Line 171 “likelihood ratio” not “likelihood”
Line 172: I think p should be stated as p-value because it’s not defined as an abbreviation here.

Results
Line 269: Please rephrase the sentence for more clarification. “similar pattern than the index-based in new cases”.
Discussion and conclusion
Line 315: what does the author mean by “founded”?
Line 355: I suggest explicitly stating this as “High population density”
Line 359: Is stating “As point by Jo et al, (2021)” necessary in this case? Please clarify.

Additional comments
Line 19: The “and” between cost and resistance distances isn’t necessary.
Line 28: Is this comma “2,70 million” required?
Line 67: The sentence is not grammatically correct, I think it’s “carrying out”.
Line 78: I think typo “number” not “numbers”. The author needn’t make this plural.
Line 148: I think you should include “which is widely used to detect clusters” to improve the grammatic flow
Line 205: Typo determine not determinate
Line 205-211: Please maintain the grammatic tense in the manuscript
Line 232: typo municipalities not minucipalities.
Line 250: I think strength is a more precise word than strongness.
Line 265: figure 5 typo: right y-axis not rigth y-axis.
Line 269: Please include the hyphen “index-based”

---

## Round 0.4 · accepted · Accept

Dear Authors,
Thank you for addressing my comments and the reviewer comments, congratulations, your manuscript is accepted.
Best wishes,
S